# Developing an observation protocol for online STEM courses

**Brian S. Horvitz**[1]*, **Whitney DeCamp**[2], **Regina Garza Mitchell**[1], **Megan Kowalske**[3], **Cherrelle Singleton**[1]

1 Department of Educational Leadership, Research & Technology, Western Michigan University, Kalamazoo, MI, United States of America, 2 Department of Sociology, Western Michigan University, Kalamazoo, MI, United States of America, 3 Department of Chemistry, Western Michigan University, Kalamazoo, MI, United States of America

* brian.horvitz@wmich.edu

**Data Availability Statement:** All relevant data are within the paper and its Supporting Information files.

**Funding:** This material is based upon work supported by the National Science Foundation

## Abstract

The use of online instruction for undergraduate STEM courses is growing rapidly. While researchers and practitioners have access to validated instruments for studying the practice of teaching in face-to-face classrooms, analogous tools do not yet exist for online instruction. These tools are needed for quality design and control purposes. To meet this need, this project developed an observational protocol that can be used to collect non-evaluative data for the description, study, and improvement of online, undergraduate STEM courses. The development of this instrument used a sequential exploratory mixed methods approach to the research, design, pilot-testing, refinement and implementation of the protocol. Pairs of researchers tested the final version of this instrument, observing completed online undergraduate STEM courses. Across 2,394 pairs of observations, the observers recorded the same indication (yes or no to the presence of some course element) 1,853 times for an agreement rate of 77.4%, falling above the 75% threshold for an acceptable level of agreement. There was a wide range in the inter-rater reliability rates among items and further revisions were made to the instrument. This foundational work-in-progress instrument should be further developed and used by practitioners who are interested in learning about and reflecting on their online teaching practice.

## Introduction

Measurement produces improvement [1], and the growing availability of instruments and tools for describing and measuring instructional practices has been valuable in promoting more effective teaching practices in face-to-face undergraduate science, technology, engineering, and mathematics (STEM) courses. Enrollment in online courses is growing much faster than higher education as a whole and accounts for a large share of undergraduate enrollment [2, 3]. Yet few research-based tools exist to help consistently measure instructional conditions in these online settings. Presently, no observational protocol for online STEM courses has been reported in the literature that has been subjected to testing and iterative improvements. In order to scale STEM instruction across all classroom modalities, systematic, valid, and

under Grant No. 1712065, received by BSH, WD, RGM, and MGK. Any opinions, findings, and conclusions or recommendations expressed in this material are those of the authors and do not necessarily reflect the views of the National Science Foundation. The funders had no role in study design, data collection and analysis, decision to publish, or preparation of the manuscript. https://www.nsf.gov/awardsearch/showAward?AWD_ID=1712065.

**Competing interests:** The authors have declared that no competing interests exist.

reliable ways to measure STEM instruction in online courses is needed. Thus, this project merges expertise in STEM education, online instruction, and educational measurement to develop an observational protocol that aids in the planning, implementation, and evaluation of online undergraduate STEM courses across all STEM disciplines to serve as a foundational work for practitioners seeking to learn about and reflect on their online teaching practice.

## Literature review

### Existing measurement instruments for STEM classrooms

Substantial seminal research has articulated how undergraduate students learn and which teaching practices best support student learning [4–6]. There are empirically validated curricula and instructional strategies for postsecondary classrooms. The effort to transform postsecondary courses to include more of these empirically validated strategies has resulted in expansive efforts to accurately describe what teaching practices actually occur in college classrooms.

Surveys of teaching practices (e.g. Faculty Survey of Student Engagement) [7] and observational instruments for classifying instructor behavior in the classroom like the Reform Teaching Observation Protocol (RTOP) [8] and the Teaching Dimensions Observation Protocol (TDOP) [9] are widely used to paint a comprehensive portrait of (a) what instructors report about their teaching and (b) what teaching practices are actually observed in postsecondary classrooms [10]. For an in-depth literature review of several more classroom observational protocols, see Guimaraes & Lima [11]. These methods in combination (observation and self-report) provide an objective portrait of postsecondary teaching that serves as a baseline for individual instructors, colleges, and faculty developers to plan and enact change initiatives, and for researchers to measure the influences of organizational factors and impacts of change initiatives on instructors' practices. Yet, they only cover a portion of the teaching and learning landscape.

Although the adoption of online learning approaches is increasing across higher education [2, 12], there is still a need for instruments that measure online teaching practices. There is also a need for an objective set of descriptors that can be used to help classify online teaching practices. For practitioners and researchers to be able to describe and evaluate online instructional practices, they will first need clear definitions of these practices. Only then can they work to improve such practices [13].

### Online learning in post-secondary education

Online enrollments have consistently grown faster than overall enrollment in higher education. By 2020, 84 percent of U.S. postsecondary students had some or all of their classes online [2]. The White House [14], as part of their plan to make college more affordable while promoting quality, has called for the proliferation of redesigned courses that blend in- person and online experiences.

Researchers and practitioners in online education have primarily focused their work on improving student outcomes. They know what factors relate to student success (e.g. attitudes about technology, motivations for completing online coursework, the amount and nature of online student-student and student-instructor interactions) and compared student learning outcomes among online, face-to-face, and blended learning [15, 16]. In comparison studies of distance and face-to-face courses, student success was not dependent on the type of technology used, but rather the instructional methods used in the course [17, 18]. We need to better understand these instructional practices to improve instruction for undergraduate students in STEM fields.

Most research about online instruction examines instructors' attitudes about educational technology, their choice to use particular platforms (Blackboard, Sakai) or tools (discussion boards, wikis, etc.), or students' perspectives of the instructor [19]. Although best practice recommendations are common [20–25], empirical research on actual use of these practices is rare. A review of the literature using multiple search terms returned only one article that examined online instructional practices [26], and this investigated how community college faculty implemented multicultural teaching approaches online. This project fills a notable gap by examining the nature of online instructional practice in STEM courses.

To help address the gap in the literature on online instructional practices, this project focused on the development of an instrument that can help researchers examine what is happening in online instructional practice. Significant effort by instructional designers, faculty developers, and online platform providers have provided checklists and rubrics of best practices including the Quality Matters Higher Education Rubric [27], BlackBoard Exemplary Course Program Rubric [28], the MERLOT (Multimedia Educational Resource for Learning and Online Teaching) Peer Review Report Form [29], and the Online Learning Consortium Quality Scorecard Suite [30]. For an in-depth review of a variety of online learning evaluation instruments, see the work of Baldwin and Trespalacios [31]. Some of these instruments have conceptual foundations in teaching practice research, such as Chickering and Gamson [4]. However, these rubrics are designed for self-reflection or for peer evaluation. They are not designed to consistently measure the same instructional practices over separate administrations (reliability), nor are they confirmed to measure what they are intended to measure (validity). For proper comparisons among data sets and accurate results, valid and reliable instruments should be designed to measure instructional practices in online settings that account for multiple dimensions of student learning in an online environment. The purpose of this project was to develop an observational protocol instrument that can aid in describing how instructors teach in online STEM courses. The following section describes our instrument development effort.

## Conceptual framework for describing online instruction

We used the Community of Inquiry (CoI) framework to guide our understanding of teaching throughout the process developing our instruments. The CoI framework acknowledges the cognitive and social dimensions of learning [32, 33]. The framework has been extensively used, as summarized in a special of The Internet and Higher Education which serves as a ten-year retrospective on its use in educational research [34, 35]. Research conducted under the CoI framework has also examined epistemic engagement in online learning [36], the effects of instructional methods on the quality of student interaction [37], and the development of community in blended learning [38].

The CoI framework suggests that deep and meaningful learning experiences are developed through three interdependent elements: social presence, cognitive presence, and teaching presence. The element of teaching presence is the focus of our work. Garrison and Arbaugh [39] define teaching presence as "the design, facilitation and direction of cognitive and social processes for the purpose of realizing personally meaningful and educationally worthwhile learning outcomes" (p. 163). The importance of teaching presence for successful online teaching has been widely supported [40–42], and is considered a key factor in student satisfaction, perceived learning, and sense of community [39].

Teaching presence has three components:

1. Instructional design and organization refers to "the planning and design of the structure, process, interaction and evaluation aspects of the online course" [39, p. 163].

2. Facilitating discourse refers to "the means by which students are engaged in interacting about and building upon the information provided in the course instructional materials" [39, p. 164].

3. Direct instruction refers to "the instructor's provision of intellectual and scholarly leadership, in part through sharing their subject matter knowledge with the students" [39, p. 164].

Teaching presence therefore examines the mechanisms through which instructors and students interact with each other and with course content. This framework is similar to previous observational protocols (e.g. RTOP, TDOP) and instructor surveys designed for face-to-face classrooms, as well as the survey developed for Henderson's NSF- WIDER project [43]. The CoI framework aligns with the constructs developed in other research and is already well established in the education research community [34, 35], making it a good fit for observing teaching practices in online STEM environments.

Although CoI is present in both extant research and in online teaching practices (see above), applying previous instruments is not necessarily possible given the differences between a traditional classroom and an online one, particularly for asynchronous courses. Observation protocols are not easily adapted from methods used for in-person classrooms and thus require research to build new instruments from the ground up. This is not to say that online teaching cannot be observed or even that it is more difficult to observe; online instruction results in observation opportunities not possible in a traditional classrooms due to the ability to access a virtual classroom remotely and discretely [13]. The preservation of student-student and student-teach communications online, as well as of instructional materials is an advantage in observing online classrooms over face-to-face classrooms.

Research in computer-supported collaborative learning (CSCL) has examined at how best to observe and study online learning. A review of CSCL methodological practices [44] explains that among the available data-sources available to researchers are text from asynchronous communication (not in real time); text from synchronous communication (in real time); audio and video communication; logs of student and instructor activity in a learning management system; artifacts of teacher preparation and student work; and student outcome data related to assessments. In particular, the availability of asynchronous and synchronous communication data records and logs of student and instructor activity makes observing an online course unique when compared to observing an in-person course.

## Instrument development and methods

To develop our observational protocol, we used an iterative process that involves collecting and analyzing rich qualitative data (observations, interviews) to explore the phenomenon prior to quantitative data collection and analysis (survey development and validation), as described previously [45]. This technique has been used previously in the development of observation protocols [43, 46], and well-suited to instrument development. Importantly, it places critical content analysis of the literature and descriptive observational data before and to purposefully inform instrument development [47, 48]. An exploratory sequential mixed methods approach allowed us to better understand the types of instructional approaches used in online teaching through providing multiple viewpoints and analysis using both subjective and objective perspectives [49]. For this reason, mixed methods approaches can provide stronger inferences [50]. This study was reviewed and approved by the Western Michigan University institutional review board. All participants provided written consent.

The instrument used for observations–the Online Observation Protocol Sheet–was developed through an iterative process, as described previously [45]. In Phase 1: Develop Set of

Constructs, a critical content analysis was conducted, four courses were observed, and four instructors were interviewed in order to develop a set of constructors that describe online teaching. In Phase 2: Develop and Test Alpha Version, the observation protocol was developed from the list of constructs. In Phase 3: Develop and Test Pilot Version, the observation protocol was pilot tested with observations in eight courses, and revised based on the observers' reported experiences and difficulties. In Phase 4: Validate Constructs, the observation protocol was field tested in ten courses. A visual representation of the four phases is included as S1 Appendix.

## Phase 1

Critical content analysis was used to organize the literature on online postsecondary STEM classrooms into the CoI teaching presence elements: instructional design and organization, facilitating discourse, and direct instruction. An organized set of elements that describes teaching in online learning environments was produced from this process, including descriptions of what can and cannot be observed. Potential observational target areas were based on the Teaching Presence element of the CoI framework [39].

As part of this process, we also (with instructor permission) observed four completed STEM undergraduate courses (including two in statistics, one in biology, and one in geosciences) and then interviewed the instructors. In order to identify potential participants for this first phase, instructors teaching online STEM undergraduate courses were identified by inspecting course schedules at a large, public, Midwestern university. Because there is no universally agreed upon list of disciplines encompassed in "STEM," we developed our own list of disciplines for purposes of recruiting instructors and courses for this study: biology, chemistry, environmental sciences, geosciences, and mathematics & statistics. Some disciplines that would fall under the STEM umbrella were not included because courses were not yet being offered online at the university used for this study. In this phase, in June 2018, ten unique instructors were identified, and personalized emails were sent to each instructor inviting them to participate in a study of STEM teaching practices in online education. Each was offered $500 for their participation, and informed that their participation would involve allowing the researchers to observe their online course materials through the school's learning management system and semi-structured interviews about their courses. In order to minimize any observation effects, observations were conducted retrospectively (i.e., only after the semester had concluded). The course subjects included one chemistry, one math, and one biology. Our team conducted the open-ended observations and semi-structured interviews for each of the courses/instructors. Data were then analyzed iteratively using a constant comparative method [51] by the project team through multi-part discussions about what is occurring, how we know what it is, and why it is important. The results of these observations and interviews were then connected to the constructs identified from the critical literature review, resulting in a tentative set of constructs and definitions to be used to fully describe online instruction. The constructs and definitions were then submitted to a panel of experts (from the project's advisory board) for expert validation. The panel individually reviewed the constructs and definitions, and provided detailed feedback for the team to use to revise for the next phase.

## Phase 2

The research team began this phase with the set of constructs produced in Phase 1 and debated which ones were observable. Those that were determined to be observable were considered for inclusion in the protocol, and items for observation were generated through a collaborative discussion by the team. The items agreed upon by the team were organized into an alpha

version of the observational protocol. The observation items were grouped into thematic sections to place similar or related elements together (e.g., quizzes near exams; textbooks near other assigned readings) to make the instrument more intuitively ordered for observers. The instrument was then reviewed and validated by the expert panel using their knowledge and experience with online courses to provide critiques regarding the extent to which the items reflect the concepts upon which they are based and whether they have face validity. The alpha version of the protocol was revised based on this feedback until no further changes were deemed necessary.

### Phase 3

The observation protocol was piloted on eight completed, online undergraduate STEM courses (with instructor permission) in May 2019. Participants were recruited using the same methods described in Phase 1, only this time participants were only asked to allow access to their courses for observations. Also, these participants were offered a $100 gift card to Amazon, Target, or Barnes & Noble for their participation. The instructors' courses in this phase included two biology, one chemistry, two geoscience, and three math/statistics courses. Different pairs of project team members were given access to each of the courses so that all eight courses would be observed with the instrument by at least two researchers. Once the observations were completed the filled-out observation protocols were compared within each individual course to see how they were filled out differently. Each pair then discussed those differences to see if those differences may have been driven by the design of protocol items. We then came to together as a team to discuss these results and used this data to make revisions to the protocol. The revised instrument was then subjected again to review by the project's expert panel.

### Phase 4

This phase required the recruitment of ten more online STEM instructors. They were recruited in the same manner as in Phase 3. Participants were recruited for this phase in March of 2021. The instructors' courses in this phase included three biology, one chemistry, one computer science, one geoscience, and four math/statistics courses. Each of the ten courses was assigned two observers from a team of five researchers. Each observer team was a unique combination, except for one pairing that was used twice. The instrument has 38 observation items recorded for each module of the course ("module" here refers to an instructor-defined unit of time and content as provided in the LMS; typically modules were about two weeks in length, but varied from course to course). Because one of the goals for the instrument is to be as easily understandable as possible, these indicators are designed to be self-explanatory to the extent possible. For this reason, as well as the large number of indicators, we did not create detailed definitions for each indicator. For ease of use by the observers, the observation items were categorized into several broad categories. Across the ten courses, there were a total of 63 modules (m = 6.3) resulting in a total of 2,394 observation points (63 modules * 38 observations per module). With two observers for each course, there were 4,788 total observations made. All observations were made and recorded independently; the observers did not discuss the course until after submitting their recorded observations.

## Results

Across the 2,394 pairs of observations, the observers recorded the same indication (yes or no) for 1,853 observation points and recorded conflicting indications for 541 observation points, resulting in an agreement rate of 77.4%. This agreement rate, which is the typical statistical indicator for interrater reliability, exceeds the 75% threshold that is considered an acceptable

level of agreement [52]. Four of the five observers agreed with each other at a rate of 74.2% to 79.6%. One observer, however, only agreed with the other observers 64.5% of the time. If that observer were treated as an outlier and excluded from the data, the remaining data would suggest 1,507 agreements out of 1,862 observations, resulting in an agreement rate of 80.9%.

The agreement rates per observation item are presented in Table 1 in the "Observer Agreement Rate" column. The agreement rates range from a low of 39.7% (formative assessments) to a high of 100.0% (audio material; instructor moderates discussion participation). A review of items with poor agreement rates and a critical analysis of the instrument with the agreement rates in mind led to a series of revisions to the instrument to potentially improve agreement rates.

First, formative and summative assessment were both among the problematic indicators. It was possible that the observers did not have compatible conceptualizations of the distinction, but the more probable issue based on observer input was that there may have been different interpretations for whether these indicators (which were within the assignments category) referred only to assignments, or also included later indicators, such as labs, discussions, and exams. It was decided to move these indicators to the end of the instrument to avoid confusion.

Second, it was unclear what counted as an "activity" and what did not. All assignments are activities in some respects, and observers seemed to each have their own opinion about what counted. Given that the assignments category already contained an "other than above" type indicator, it was determined that the activities category was redundant and thus deleted. The group work indicator was moved a new category at the end of the instrument, similar to how assessments were handled.

In addition, other edits–such as adding new examples or rewording indicators–were made throughout to address other differences in perspectives among the observers. After making these adjustments, the observers reached consensus that the revised instrument will reduce conceptual confusion about the observation indicators. The revised instrument [53] is included as S2 Appendix, and may be freely used under the Creative Common licensing (CC BY-NC-SA 4.0). A user guide is available online and may also be freely used under the Creative Common licensing [54].

## Discussion

Overall, the research team found the overall agreement rate of 77.4% across all instrument items satisfactory as it falls above the 75% threshold for an acceptable level of agreement [52]. However, as described above, there is a wide range in the inter-rater reliability rates among items within the instrument. Ideally, we would advocate continued development on the design of each of the fourteen items in the most recent version of the instrument who inter-rater reliability fell below 75%. It is also possible that some of those items with particularly low rates may be candidates for exclusion from the instrument. As described above, during the final development phase of this project, only ten courses were able to be observed. Before we would be comfortable making any claims as to the overall validity or reliability of this instrument, we recommend the survey be revised again and tested in a larger number of completed online, undergraduate STEM courses. All of the observations conducted as part of this project were conducted by the same group of five observers. It would also be helpful to use different sets of observers who are not already familiar with the instrument as that would better simulate actual use of the instrument by a practitioner or researcher who is using it for the first time.

As described above, the teaching presence element of the CoI framework has three components [39]: instructional design and organization, facilitating discourse, and direct instruction.

**Table 1. Agreement rates by item.**

| | Indicators | Observer Agreement Rate |
|---|---|---|
| General | Posted news/updates/announcements (e.g., posting a notification regarding updated grades) | 58.7% |
| | Posted guidelines for communication (e.g., guidelines for collaborative discussions or for communicating with the instructor) | 81.0% |
| | Communicates course/module goals (e.g., list student learning outcomes) | 76.2% |
| Course Materials | Assigned textbook or book | 65.1% |
| | Course pack | 87.3% |
| | Lecture notes or slides | 82.5% |
| | Other text-based materials | 61.9% |
| | Images or illustrations | 63.5% |
| | Slides with audio narration | 42.9% |
| | Audio material (other than slides) | 100.0% |
| | Video material | 61.9% |
| Assignments | Written assignments (e.g., a written assignment can be a short-answer question, essay, or research paper) | 81.0% |
| | Math problems or nomenclature (e.g., a problem set featuring math or chemistry problems) | 71.4% |
| | Problem-solving scenarios (e.g., trying to find solutions to real-world problems in assignments) | 81.0% |
| | Student project | 88.9% |
| | Student presentations (e.g., synchronous video or submitted media to present or discuss findings) | 96.8% |
| | Required/graded assignments other than the above (except for activities and labs) | 63.5% |
| | Instructor provides examples for assignments | 74.6% |
| | Formative assessments (i.e., non-graded or lower risk assignments that provide immediate feedback to monitor student progress) | 39.7% |
| | Summative assessments (i.e., graded, higher risk assignments that evaluate student progress) | 55.6% |
| | Non-required/ungraded assignments | 87.3% |
| Activities | Individual activities (i.e., students are not allowed to work or collaborate with other students when completing these activities) | 65.1% |
| | Group activities (i.e., students are encouraged to work or collaborate with other students when completing these activities) | 90.5% |
| | Instructor provides a model or example for activities or assignments (e.g., rubrics) | 77.8% |
| Lab work | Laboratory assignments (e.g., using a laboratory kit or list) | 84.1% |
| | Laboratory kit or list (e.g., physical materials or a list of physical items to acquire) | 98.4% |
| | Simulation/visualization website or software | 68.3% |
| | Synchronous video labs (e.g., meeting with the class through video conferencing in real-time) | 93.7% |
| | Video (synchronous or asynchronous) of student work (e.g., submitting a step-by-step process of solving a mathematical problem) | 93.7% |
| | Physical or virtual models | 82.5% |
| Discussion | Discussion forums (e.g., student-student discussions) | 92.1% |
| | Instructor moderates participation of students in the discussion forum (e.g., draws out inactive students/limits dominating students) | 100.0% |
| | Instructor contributes to discussion (e.g., adds information, builds consensus, summarizes, diagnoses misconceptions, provides encouragement, etc.) | 87.3% |
| | Synchronous video discussions | 76.2% |

(*Continued*)

**Table 1.** (Continued)

| | Indicators | Observer Agreement Rate |
|---|---|---|
| Testing | Ungraded/practice quizzes/exams | 79.4% |
| | Quizzes (e.g., weekly or bi-weekly assessments) | 82.5% |
| | Exams (e.g., end of unit assessments, mid-term exams, etc.) | 79.4% |
| | Review of quiz/exams answers (e.g., which questions a student got right or wrong and/or correct answers) | 69.8% |
| Overall Agreement Rate | | 77.4% |

Through the four phases of this instrument development process, particularly during the critical content analysis process, instrument items were created that reflect each of these three components. The use of expert panel review in phases 1, 2 and 3 along with the overall observer agreement rate of 77.4% (with variation between sets of observers, as described above) provide evidence that the final version of the observational protocol will give its users, be they practitioners or researchers, online teaching data that reflects key components of the CoI framework. The efficacy of such an approach that combines a variety of research methods with the aim of increasing the new instrument's validity is supported by relevant methodological literature [47–49].

## Limitations

As is unavoidable in the production of any instrument, the instrument developed here is subject to bias and limitations as a result of the researchers who developed it and the underlying framework. The research team included expertise in chemistry, higher education leadership, instructional technology, research methods, sociology, and other subjects relevant to this work, but this is by no means an exhaustive set of expertise that covers the subjects and teaching approaches used in courses within the scope of this project. Efforts were made to include a wide range of subjects with which to field test the instrument, but further use and evaluation is appropriate to refine the instrument and minimize biases, and to ensure that the instrument is compatible with subjects beyond those in the samples used here.

## Conclusions

Although we do advocate for continued work on this observational protocol to increase its item and overall reliability, we want to reemphasize that no other observational protocol for online STEM courses has been reported in the literature, let alone been subjected to this degree of testing, revision, and re-testing. This current version of the instrument is therefore a foundational work in progress that may not be ready for use as a research instrument without further testing, but could certainly be useful by practitioners who are interested in learning about and reflecting on their online teaching practice.

## Supporting information

**S1 Data.**
(CSV)

**S1 Appendix. Observation instrument design process.**
(TIF)

**S2 Appendix. Online observation protocol sheet.**
(ZIP)

## Acknowledgments

The authors would like to thank the instructors who participated in this study, the project advisory board, and the staff members who helped facilitate this project.

## Author Contributions

**Conceptualization:** Brian S. Horvitz, Whitney DeCamp, Regina Garza Mitchell, Megan Kowalske.

**Data curation:** Brian S. Horvitz, Whitney DeCamp, Regina Garza Mitchell, Megan Kowalske.

**Formal analysis:** Brian S. Horvitz, Whitney DeCamp, Regina Garza Mitchell, Megan Kowalske, Cherrelle Singleton.

**Funding acquisition:** Brian S. Horvitz, Whitney DeCamp, Regina Garza Mitchell, Megan Kowalske.

**Investigation:** Brian S. Horvitz, Whitney DeCamp, Regina Garza Mitchell, Megan Kowalske, Cherrelle Singleton.

**Methodology:** Brian S. Horvitz, Whitney DeCamp, Regina Garza Mitchell, Megan Kowalske, Cherrelle Singleton.

**Project administration:** Brian S. Horvitz.

**Resources:** Brian S. Horvitz, Whitney DeCamp, Regina Garza Mitchell, Megan Kowalske, Cherrelle Singleton.

**Software:** Whitney DeCamp.

**Supervision:** Brian S. Horvitz.

**Validation:** Brian S. Horvitz, Whitney DeCamp, Regina Garza Mitchell, Megan Kowalske, Cherrelle Singleton.

**Writing – original draft:** Brian S. Horvitz, Whitney DeCamp, Regina Garza Mitchell, Megan Kowalske.

**Writing – review & editing:** Brian S. Horvitz, Whitney DeCamp, Regina Garza Mitchell.

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
