## [Decision Letter · Decision Letter 0]

5 Jul 2023

PONE-D-23-10329Developing an Observation Protocol for Online STEM CoursesPLOS ONE

Dear Dr. Horvitz,

Thank you for submitting your manuscript to PLOS ONE. After careful consideration, we feel that it has merit but does not fully meet PLOS ONE’s publication criteria as it currently stands. Therefore, we invite you to submit a revised version of the manuscript that addresses the points raised during the review process.

We look forward to receiving your revised manuscript.

Kind regards,

Dr Daner Sun

Academic Editor

PLOS ONE

Journal Requirements:

"The authors would like to thank the National Science Foundation, the instructors who participated in this study, the project advisory board, and the staff members who helped facilitate this project."

"This was research was funded by a grant from the U.S National Science Foundation. All authors (BH, WD, RGM, MK, CS) were part of this award.

URL: https://www.nsf.gov/div/index.jsp?div=DUE

The funders did not play any role in this work beyond funding it. "

"No authors have competing interests."

Additional Editor Comments:

Please see reviewer's comments

Reviewers' comments:

Reviewer's Responses to Questions

**Comments to the Author**

1. Is the manuscript technically sound, and do the data support the conclusions?

Reviewer #1: Yes

Reviewer #2: Yes

2. Has the statistical analysis been performed appropriately and rigorously? 

Reviewer #1: Yes

Reviewer #2: Yes

3. Have the authors made all data underlying the findings in their manuscript fully available?

Reviewer #1: Yes

Reviewer #2: Yes

4. Is the manuscript presented in an intelligible fashion and written in standard English?

Reviewer #1: Yes

Reviewer #2: Yes

5. Review Comments to the Author

Reviewer #1: The manuscript presents a sequential mixed methods approach that was used to design, pilot-test, refine, and implement an observation protocol. The key contribution of this research is the development of a validated instrument that can collect non-evaluative data in online STEM courses. However, there are some major flaws that need to be addressed before accepting this manuscript.

1. Abstract: the implications of the study should be stated in the abstract. “The research team recommends reconsidering the inclusion of some items with low levels of agreement or continued revisions on such items.” What does this sentence refer to? Please elaborate it.

2. Introduction: The introduction cannot adequately justify this study.

In the Introduction of a research paper, authors usually include the background, purpose, scope, and significance of this research, as well as the need for conducting this study.

The introduction establishes the importance and relevance of the project by highlighting the gap in measuring instructional practices in online STEM courses. However, the current introduction was too short.

I suggest that it could be improved in several ways. First, it could cite more recent sources to support the claims about the growth of online STEM courses, as well as the lack of accurate methods to measure online instruction. Second, the authors may define “accurate” in this study. “Accurate” is a very subjective and ambiguous word. it could clarify what constitutes an observational protocol and how it differs from other instruments or tools for describing and measuring instructional practices. Third, it could specify the intended audience and scope of the project, such as whether it focuses on a particular discipline, level, or type of online STEM course.

3. Literature review:

3.1 The logic in the “Existing Measurement Instruments for STEM Classrooms” was loose. For example, the authors listed some instruments (face-to-face instruments) in the first and second paragraph. Then, in the third paragraph, the authors stated, “Although the adoption of online learning approaches is increasing across higher education (Johnson, Adams, Becker, Estrada, & Martín, 2013), there is still a need for instruments that measure online teaching practices.” In my humble opinion, I cannot see a strong connection between paragraph 2 and paragraph 3. The authors did not introduce the current online instruments for STEM courses.

I suggest a comprehensive review of instruments for teaching practices in both traditional classrooms and online courses are needed.

3.2 In addition, the literature should be updated. Some selected studies are not the latest.

3.3 there are some confusing terms need further explanation. For example, the second paragraph on page 4, “Significant effort by instructional designers, faculty developers, and online platform providers have provided checklists and rubrics of best practices (e.g., Quality Matters Higher Education Rubric, BlackBoard Exemplary Course Program Rubric, MERLOT Evaluation Standards for Learning Materials, Online Learning Consortium

Quality Scorecard). The terms in parentheses need to be explained, because they are difficult to understand for readers who see these terms for the first time.

3.4 The framework of CoI. Section of “Conceptual Framework for Describing Online Instruction” should be improved.

3.4.1. On Page 4, the authors should provide a clear definition of what they mean by effective online teaching practices and how they operationalize them in their instruments. They should also explain how the CoI framework aligns with their definition and operationalization of effective online teaching practices.

3.4.2. A more recent and relevant literature that supports the validity and reliability of the CoI framework for measuring online teaching practices are strongly needed. The authors should justify why they chose to use the CoI framework over other frameworks or models that have been proposed for online teaching practices. They should also discuss how their instruments address the gaps or challenges that have been identified in previous studies using the CoI framework.

3.4.3. The authors should avoid making general claims or implications based on the CoI framework without providing empirical evidence or data from their own study or other studies. They should also acknowledge the limitations and potential biases of their instruments and the CoI framework. The reason of “the framework has been extensively used, including in two special issues of the Internet and Higher Education in 2010” are not convincing.

3.4.4. The last paragraph (p6) in the literature review section should be put in “Methods”. The reasons of choosing mixed research methods should be further elaborated.

4. Instrument development and methods:

The methodology section is well-written and provides a detailed description of the sequential mixed methods approach used to develop the observation protocol. However, it would be helpful to provide more information on the selection criteria for the online STEM courses observed and the characteristics of the observers.

4.1 I suggest adding a figure showing the iterative process.

4.2. “four courses were observed”, what are the four courses? Detailed information is needed.

4.3. Do authors think the subjects in STEM courses will influence the observation items?

5. Results

The results section presents the agreement rates among observers for each item in the observation protocol. It is expected that each indicator could be explained in the section 4. What does these indicators refer to in the STEM courses.

6. Discussion

The discussion section provides a thoughtful analysis of the implications of the results for the future design and control of online STEM courses. However, it would be helpful to provide more information on how the protocol aligns the framework of CoI and how the current study relates to previous studies. Literature should be cited in the discussion section.

Overall, I believe that your manuscript makes a valuable contribution to the field of online STEM education and the development of validated instruments for studying and improving online instruction. I recommend that you address the above comments and revise your manuscript accordingly.

Reviewer #2: The study proposed an analogous tool for online instruction of undergraduate STEM courses. The development of this instrument used a sequential mixed methods approach with 4 phases, i.e. to the research (critical content analysis), design (developing the list of constructs), pilot-testing, refinement and implementation of the protocol (field testing in ten courses).

Overall all speaking, the study is worthwhile and provides useful instruments for educators to develop their online STEM courses, or to evaluation their courses through self-refection. The instruments can be a set of checklists for course developers’ reference too.

However, the article seldom mentioned the reason why the items were grouped in 7 criteria, namely General, Course Materials, Assignments, Activities, Lab work, Discussion and Testing. More literature can be added to support the use of the 7 criteria.

Besides, the four phases were not clearly defined in the article. In the abstract, they were put as “research, design, pilot-testing, refinement and implementation of the protocol.” In Line 163 to 216, the authors use different terms to explain them. It is advised to align with the team and clearly define the 4 phases. The authors may use a figure to illustrate this iterative process.

6. PLOS authors have the option to publish the peer review history of their article (what does this mean?). If published, this will include your full peer review and any attached files.

Reviewer #1: **Yes: **Yin YANG

Reviewer #2: No

---

## [Author Response · Author response to Decision Letter 0]

15 Sep 2023

Please see our responses to the reviewers' comments in the Response to Reviewers document uploaded.

---

## [Decision Letter · Decision Letter 1]

4 Dec 2023

PONE-D-23-10329R1Developing an Observation Protocol for Online STEM CoursesPLOS ONE

Dear Dr. Horvitz,

Thank you for submitting your manuscript to PLOS ONE. After careful consideration, we feel that it has merit but does not fully meet PLOS ONE’s publication criteria as it currently stands. Therefore, we invite you to submit a revised version of the manuscript that addresses the points raised during the review process.

We look forward to receiving your revised manuscript.

Kind regards,

Daner Sun

Academic Editor

PLOS ONE

Journal Requirements:

Additional Editor Comments:

The authors have well addressed the comments.

Reviewers' comments:

Reviewer's Responses to Questions

**Comments to the Author**

1. If the authors have adequately addressed your comments raised in a previous round of review and you feel that this manuscript is now acceptable for publication, you may indicate that here to bypass the “Comments to the Author” section, enter your conflict of interest statement in the “Confidential to Editor” section, and submit your "Accept" recommendation.

Reviewer #3: (No Response)

Reviewer #4: (No Response)

2. Is the manuscript technically sound, and do the data support the conclusions?

Reviewer #3: Yes

Reviewer #4: Yes

3. Has the statistical analysis been performed appropriately and rigorously? 

Reviewer #3: Yes

Reviewer #4: Yes

4. Have the authors made all data underlying the findings in their manuscript fully available?

Reviewer #3: Yes

Reviewer #4: Yes

5. Is the manuscript presented in an intelligible fashion and written in standard English?

Reviewer #3: Yes

Reviewer #4: Yes

6. Review Comments to the Author

Reviewer #3: 1. Row 34, please notice the use of brackets; Row 100, please notice the format. Row 203, add “,” after “completed”. Row 212, “each of the ten courses was”.

2. Please notice the concept of teaching presence when talking about the CoI framework. The description here, in rows 122 to 123, “instructors and students interact with each other and with course content,” is more connected to teaching presence and social presence. If social presence is also an important concept for the research?

3. The elements of teaching presence from 169–170 may be deleted to make this part clearer.

4. Can the number of participants or observations of the courses in phrase 4 be informed?

5. In the result part, can statistic analysis, for example, the chi-square test or t test, or other methods be used to examine the reliability or consistency of the results of two observers?

6. The section “Introduction” should be more powerful to emphasize the importance and significance of the development of the observation protocol.

7. There should be more connection with the former research in the section "Discussion". It may be from the perspective of, for example, the validation of the design or processing.

8. Content from rows 275 to 287 may be in the section “limitations and future”.

9. After the construction of the observation protocol, it should be displayed in the main body, and each section should be explained. Although the protocol is mainly developed from the CoI framework, the origin of each section should be referenced and explained well.

10. The definition of “STEM course” is different from science subject courses, such as math, geoscience, and biology. Thus, please clarify the definition of “STEM course” in the section “Introduction” or “Literature Review”. At the same time, please check if the references related to STEM courses or STEM education are correctly referenced.

Reviewer #4: The article has made general adjustments in response to the reviewers' suggestions. However, a few minor issues still need attention. First, it is recommended that the writing of the Review of Literature section be revised to Literature Review. Also, the Conclusion section was not found in the article, so it is suggested that a Conclusion section be added, or that the two be combined to change the subtitle to Conclusion and Discussion. Furthermore, there are some editorial concerns within the article, so it is kindly requested that the article format be carefully reviewed and refined.

7. PLOS authors have the option to publish the peer review history of their article (what does this mean?). If published, this will include your full peer review and any attached files.

Reviewer #3: **Yes: **ZHENG Zhizi

Reviewer #4: No

---

## [Author Response · Author response to Decision Letter 1]

7 Dec 2023

Please see the attached document, "Response to Reviewers 120723" in which we detail our responses to each of the reviewers' comments.

---

## [Editor Report · Decision Letter 2]

4 Jan 2024

Developing an Observation Protocol for Online STEM Courses

PONE-D-23-10329R2

Dear Dr. Horvitz,

We’re pleased to inform you that your manuscript has been judged scientifically suitable for publication and will be formally accepted for publication once it meets all outstanding technical requirements.

Kind regards,

Daner Sun

Academic Editor

PLOS ONE

---

## [Editor Report · Acceptance letter]

17 Jan 2024

PONE-D-23-10329R2 

PLOS ONE

Dear Dr. Horvitz, 

I'm pleased to inform you that your manuscript has been deemed suitable for publication in PLOS ONE. Congratulations! Your manuscript is now being handed over to our production team.

Kind regards, 

on behalf of

Dr. Daner Sun 

Academic Editor

PLOS ONE